# An Angelman syndrome substitution in the HECT E3 ubiquitin ligase *C*-terminal Lobe of E6AP affects protein stability and activity

**Steven A. Beasley, Chloe E. Kellum, Rachel J. Orlomoski, Feston Idrizi, Donald E. Spratt** *

Gustaf H. Carlson School of Chemistry and Biochemistry, Clark University, Worcester, MA, United States of America

* dspratt@clarku.edu

**Data Availability Statement:** All relevant data are within the manuscript. All chemical shift assignments and experiments were deposited into the Biological Magnetic Resonance Databank

## Abstract

Angelman syndrome (AS) is a rare neurodevelopmental disorder characterized by speech impairment, intellectual disability, ataxia, and epilepsy. AS is caused by mutations in the maternal copy of *UBE3A* located on chromosome 15q11-13. *UBE3A* codes for E6AP (*E6 Associated Protein*), a prominent member of the HECT (*Homologous to E6AP C-Terminus*) E3 ubiquitin ligase family. E6AP catalyzes the posttranslational attachment of ubiquitin *via* its HECT domain onto various intracellular target proteins to regulate DNA repair and cell cycle progression. The HECT domain consists of an *N*-lobe, required for E2~ubiquitin recruitment, while the C-lobe contains the conserved catalytic cysteine required for ubiquitin transfer. Previous genetic studies of AS patients have identified point mutations in *UBE3A* that result in amino acid substitutions or premature termination during translation. An AS transversion mutation (codon change from ATA to AAA) within the region of the gene that codes for the catalytic HECT domain of E6AP has been annotated (I827K), but the molecular basis for this loss of function substitution remained elusive. Here, we demonstrate that the I827K substitution destabilizes the 3D fold causing protein aggregation of the *C*-terminal lobe of E6AP using a combination of spectropolarimetry and nuclear magnetic resonance (NMR) spectroscopy. Our fluorescent ubiquitin activity assays with E6AP-I827K show decreased ubiquitin thiolester formation and ubiquitin discharge. Using 3D models in combination with our biochemical and biophysical results, we rationalize why the I827K disrupts E6AP-dependent ubiquitylation. This work provides new insight into the E6AP mechanism and how its malfunction can be linked to the AS phenotype.

## Introduction

Angelman syndrome (AS) is a neuro-genetic disorder that effects 1 in 15,000 people characterized by symptoms such as developmental delay, speech impairment, intellectual disability, walking and balance disorders, and epilepsy [1–3]. Individuals also demonstrate a unique behavioral pattern that typically includes a happy demeanor, easily provoked laughter, short

(http://www.bmrb.wisc.edu) under accession code 50084.

**Funding:** This work was supported by the National Institutes of Health (R15GM126432 to D.E.S.; www.nigms.nih.gov) and start-up funds from Clark University (D.E.S.; www.clarku.edu). The funders did not play any role in the study design, data collection and analysis, decision to publish, or the preparation of the manuscript.

**Competing interests:** The authors have declared that no competing interests exist.

attention span, sleep disturbance, and an affinity for water [2, 4]. AS is caused by the loss of gene function of the maternal copy of *UBE3A* on chromosome 15. *UBE3A* is paternally imprinted in neurons resulting in expression of the maternal allele alone, whereas other tissues retain normal biallelic expression patterns [2, 5–8]. The Human Gene Mutation Database (HGMD) currently lists 161 different genetic mutations of *UBE3A* [9]. Of the genetic mechanisms that have been described as the cause of AS, an estimated 70–80% of cases contain deletions in the maternal chromosome 15q11-q13 [2, 3]. Another 10–20% of affected individuals harbor mutations in their maternally inherited *UBE3A* gene, 3–5% of cases are due to two paternal copies of the chromosome, and lastly 3–5% of affected individuals have the paternal imprint of the maternal chromosome leaving no functioning copy of the *UBE3A* gene [2, 3].

*UBE3A* codes for the E3 ubiquitin ligase E6-Associated Protein (E6AP). This protein is a member of the Homologous to E6AP Carboxy-Terminus (HECT) family of E3 ubiquitin ligases and plays an important role in the ubiquitylation-signaling pathway. Ubiquitylation is a post-translational modification of targeted proteins that initiates processes such as protein degradation, intracellular trafficking, and other signaling events [10, 11]. Ubiquitin is transferred onto a substrate through a cascade of three enzymes (E1, E2, E3), with the combinatorial effect of the approximately 40 E2 and over 600 E3 enzymes ultimately determining substrate specificity [12–14]. The fate of the ubiquitylated substrate is determined by the ubiquitin linkage chain type by the E2/E3 combination [10, 11, 15]. For example, K29 and K48 linkages target a substrate protein for proteasomal degradation, while K63 linkages are involved in DNA repair mechanisms and intracellular targeting [10, 11, 16]. The dysregulation of any of these components involved in the ubiquitylation process leads to a myriad of different diseases including various cancers, developmental disorders, and neurodevelopmental disorders including AS [8, 16–18].

E6AP was first identified as an E3 ubiquitin ligase through its ability to target the tumor suppressor protein p53 for degradation in conjunction with the human papilloma virus protein E6 [19, 20]. E6AP is a 100 kDa protein that catalyzes the covalent attachment of ubiquitin to its various substrates through the use of its C-terminal HECT domain (residues 518–875). The HECT domain consists of an N-terminal lobe (N-lobe) and a C-terminal lobe (C-lobe) connected by a flexible three-residue hinge [21]. A broad cleft at the interface of the two lobes contains the catalytic cysteine required for ubiquitylation [21]. E6AP selectively builds K48-polyubiquitination chains, consistent with its ability to target substrates for proteasomal degradation, and this ubiquitin chain-linkage specificity is located in the C-lobe of the E6AP HECT domain [22]. E6AP has been shown to interact with numerous cellular proteins and can regulate a number of different homeostatic cellular processes [23]. Prime examples of E6AP-regulated processes include cell cycle control through centrosomal regulation [24], DNA repair through its interaction with UV excision repair protein RAD23 homolog A (HHR23A) [25], targeting tuberin (TSC2) for proteosomal degradation [26], signal transduction by binding to multiple Src family tyrosine kinases [27], breast cell proliferation through calmodulin/$Ca^{2+}$ mediated proteosomal degradation of the estrogen receptor (ER) [28], and coordinating the inflammatory response in conjunction with annexin A1 [29].

While the genetic link between mutations in *UBE3A* and AS is well established in the literature [1–4, 6, 7], the effect that each specific AS mutation has on the translated protein product have not been fully characterized or well understood. In this study, we show that the AS I827K substitution (also described as I804K based on an alternate open reading frame start codon [30–33]) partially disrupts the overall 3D fold of the HECT C-terminal lobe of E6AP leading to its aggregation and diminished E6AP-ubiquitylation activity *in vitro*. We unambiguously demonstrate that the AS I827K mutation in *UBE3A* is a loss of function mutation. This biophysical

study clarifies how the I827K substitution in the C-terminal lobe domain of E6AP contributes to the Angelman syndrome phenotype.

## Methods and materials

### Cloning and site-directed mutagenesis

The original DNA construct for the human HECT C-lobe of E6AP (E6AP$^{C\text{-lobe}}$; Uniprot Q05086, residues 761–875) was codon optimized and synthesized by ATUM (Newark, CA, USA) [34–36]. The gene was cloned into an ampicillin-resistant T7-inducible plasmid with an N-terminal His$_6$ affinity tag followed by a TEV protease cleavage site (ENLYFQ/GS). The resulting His$_6$-TEV-E6AP$^{C\text{-lobe}}$ vector was subsequently used as the template to insert the I827K mutation using the SPRINP protocol [37]. The catalytic cysteine was changed to an alanine (C843A) using the same protocol. Due to subsequent precipitation issues of the I827K substituted construct and to increase solubility, the E6AP$^{C\text{-lobe}}$-I827K open reading frame was subcloned into an expression vector with an N-terminal His$_6$-SUMO fusion tag using compatible 5' *Bam*HI and 3' *Xho*I restriction sites. The HECT domain of E6AP (residues 518–875) was PCR amplified from a plasmid coding for the full-length E6AP purchased from Addgene (Plasmid #8655; Watertown, MA, USA) [27] and subcloned into the His$_6$-SUMO vector using compatible 5' *Bam*HI and 3' *Xho*I sites. All plasmids (pHis$_6$-TEV-E6AP$^{C\text{-lobe}}$, pHis$_6$-SUMO-E6AP$^{C\text{-lobe}}$-I827K, pHis$_6$-SUMO-E6AP$^{HECT}$ and pHis$_6$-SUMO-E6AP$^{HECT}$-I827K) were isolated using the Monarch Plasmid Miniprep kit (New England Biolabs, Ipswich, MA, USA), quantified by A$_{280}$ using a Nanodrop One$^C$ UV-Vis spectrophotometer (Thermo-Fisher, Waltham, MA, USA), and verified by DNA sequencing (Macrogen, Cambridge, MA, USA).

### Protein expression and purification

The pHis$_6$-TEV-E6AP$^{C\text{-lobe}}$, pHis$_6$-SUMO-E6AP$^{C\text{-lobe}}$-I827K, pHis$_6$-SUMO-E6AP$^{HECT}$ and pHis$_6$-SUMO-E6AP$^{HECT}$-I827K expression plasmids were transformed into *E. coli* BL21 (DE3) RIL+ competent cells and grown at 37°C in Luria-Bertani media supplemented with ampicillin 100 mg/L and chloramphenicol 34 mg/L. The E6AP proteins grown for heteronuclear NMR analysis were grown in minimal M9 media (2 x 1L) supplemented with 1 g/L of $^{15}$NH$_4$Cl and 2 g/L of $^{13}$C-glucose as the sole nitrogen and carbon sources. When the cultures reached an OD$_{600}$ of 0.6–0.8, protein expression was induced with the addition of 0.5 mM IPTG for 20 hours at 16°C. The cells were harvested by centrifugation 6000 x g for 10 minutes at 4°C using a Sorvall LYNX 4000 superspeed centrifuge with a Fiberlite F10-4x1000 LEX Carbon Fiber rotor (Thermo-Fisher) and resuspended in cold wash buffer (50 mM Na$_2$HPO$_4$ pH 8.0, 300 mM NaCl, 10 mM imidazole) supplemented with ProBlock Gold Bacterial Protease inhibitor cocktail (GoldBio, St. Louis, MO, USA). The cells were then lysed using an Avestin EmulsiFlex-C5 Homogenizer (Avestin, Ottawa, ON, Canada) and clarified by ultracentrifugation using an Optima L-80 XP ultracentrifuge with a Ti 70.1 rotor (Beckman-Coulter) for 40 minutes at 41,000 rpm at 4°C. The clarified supernatant containing the desired His$_6$-tagged E6AP protein was then isolated using 5 mL of HisPur Ni-NTA resin (Thermo-Fisher) and eluted with elution buffer (50 mM Na$_2$HPO$_4$ pH 8.0, 300 mM NaCl, 250 mM imidazole). Fractions containing E6AP protein were pooled and incubated at 25°C for one hour in the presence of TEV or SUMO protease to cleave the *N*-terminal His$_6$ or His$_6$-SUMO tag, followed by overnight dialysis at 4°C against wash buffer to remove excess imidazole. To separate the cleaved His$_6$- or His$_6$-SUMO tag and His$_6$-tagged protease from the desired E6AP protein, the cleaved sample was passed through the HisPur Ni-NTA resin column a second time and the flow-through containing the desired tag-free E6AP protein was collected. The E6AP protein

was then concentrated using a 10 MWCO Amicon Ultra-15 Centrifugal Filter (Millipore) to about 1 mL, and run through a Superdex-75 Gel Filtration Column using an AKTA Pure 25L Fast Performance Liquid Chromatography (FPLC) system with gel filtration buffer (50 mM HEPES, 100 mM NaCl, 1 mM DTT, pH 7.5 at 4˚C) at a flow rate of 1 mL/min. Due to inherent insolubility issues and after numerous attempts, the E6AP^HECT and E6AP^C-lobe-I827K proteins were unable to be passed through the size exclusion column.

## Circular dichroism spectroscopy

Circular dichroism spectroscopy was performed on the E6AP^C-lobe and E6AP^C-lobe-I827K substituted protein using a JASCO J-815 CD spectropolarimeter. The proteins were prepared by buffer exchange into low salt (10 mM $Na_2HPO_4$ pH 7.4, 30 mM NaCl) using a 10 MWCO Slide-A-Lyzer Dialysis Cassette (Thermo-Fisher). The samples were diluted to 70 μM and loaded into a quartz cuvette with a 1 mm pathlength. Wavelength scans were averaged from six trials recorded from 260 to 195 nm at 10˚C using 1 nm increments and an averaging time of 1 second. The same sample was used afterwards to obtain a melting curve of the mutant monitoring changes in α-helical content at 222 nm over a temperature range of 5–90˚C, temperature slope 1˚C/min, data pitch 0.3, response 4 seconds, bandwidth 1 nm, sensitivity standard 100 mdeg, and voltage of 600 mV.

## Ubiquitin activity assays

Ubiquitination assays were conducted with 10 μM Alexa Fluor 647 N-terminally labeled ubiquitin, 10 μM E1 activating enzyme UBE1 (Uba1), 15 μM E2 conjugating enzyme UBE2L3 (UbcH7), and E3 ligase (35 μM E6AP^C-lobe or E6AP^HECT, as well as their variants), 2 μM DTT, 20 mM ATP, 40 mM $MgCl_2$ in 50 mM HEPES pH 7.5, 100 mM NaCl. Each reaction was incubated at 37˚C in a water bath for 30 minutes. To determine the presence of ubiquitin~thioester intermediates, appropriate samples were supplemented with 10 mM DTT. Reactions were terminated by adding gel loading dye and heating at 95˚C in a dry bath for one minute. The samples were then loaded onto a Bis-Tris gel at pH 6.4 and run for 1 hour at 120 V. The gels were removed from the apparatus and immediately visualized on an iBright FL1000 imaging system (Thermo-Fisher) using the fluorescent gel imaging setting for Alexa Fluor 647.

## Heteronuclear NMR spectroscopy

The $^1H$-$^{15}N$ heteronuclear single quantum correlation (HSQC) spectra [38] of $^{15}N$-labeled E6AP^C-lobe (3.5 mM) and E6AP^C-lobe I827K (188 μM) were collected at 25˚C in a Varian Inova 600 MHz 4-channel solution-state NMR spectrometer equipped with a 5-mm PFG triple-resonance probe housed and maintained in the Carlson School of Chemistry and Biochemistry at Clark University. The samples were prepared to 600 μL in NMR buffer (20 mM $Na_2HPO_4$ pH 7.0, 100 mM NaCl, 2 mM TCEP, 1 mM EDTA), 10% $D_2O$. The spectra were referenced to the methyl peaks of 2 mM 4,4-dimethyl-4-silapentane-1-sulfonic acid (DSS) set at 0 ppm and 2 mM imidazole was added as an internal pH indicator [39]. The backbone resonances were sequentially assigned using standard 2D and 3D experiments from the Varian Biopack including $^1H$-$^{15}N$-HSQC, HNCACB [40], CBCA(CO)NH [41], HNCA [42–44], HN(CO)CA [44, 45], HN(CA)CO [46], and HNCO [42–44]. Side chain assignments were determined using C (CO)NH [47], H(CCO)NH [47, 48], as well as both aliphatic and aromatic [49], HCCH-TOCSY [50, 51] and $^1H$-$^{15}N$ and $^1H$-$^{13}C$-NOESY experiments [52–54]. All data were processed using NMRPipe and NMRDraw [55] and the spectra were analyzed using NMRViewJ [56, 57]. All chemical shift assignments and experiments were deposited into the Biological Magnetic Resonance Databank (http://www.bmrb.wisc.edu) under accession code 50084.

## Results and discussion

### The I827K Angelman syndrome substitution destabilizes E6AP$^{\text{C-lobe}}$

Wild-type E6AP$^{\text{C-lobe}}$ and E6AP$^{\text{C-lobe}}$-I827K Angelman syndrome substituted protein showed similar secondary structural content by circular dichroism, with minima at 208 and 222 indicating both proteins are predominantly α-helical at 10˚C (Fig 1A). To test the thermal stability of wild-type E6AP$^{\text{C-lobe}}$ and E6AP$^{\text{C-lobe}}$-I827K, melting curves were obtained for each protein from 5–90˚C. The melting curve showed a marked decrease in thermal stability of the E6AP$^{\text{C-lobe}}$-I827K ($T_m$ of 47.7˚C) when compared to E6AP$^{\text{C-lobe}}$ wild-type ($T_m$ of 57.7˚C) demonstrating that the Angelman syndrome I827K substitution appears to partially destabilize the 3D fold of the protein (Fig 1B). The Angelman mutant appears to start unfolding around 40˚C, close to the physiological temperature of 37˚C, whereas the E6AP$^{\text{C-lobe}}$ wild-type begins to unfold around 55˚C. These results indicate that the AS substitution has a similar global fold at cooler temperatures similar to the wild-type, whereas the melting curve suggests that the I827K AS substitution is detrimental to E6AP$^{\text{C-lobe}}$ stability.

### I827K Angelman syndrome substitution decreases E6AP ubiquitylation

The transfer of ubiquitin through the ubiquitylation pathway onto the catalytic cysteine of the E6AP was assessed using a fluorescent ubiquitin activity assay. Using this assay, we are able to observe the sequential transfer of fluorescent ubiquitin from the E1 activating enzyme UBE1 (aka Uba1), to the E2 conjugating enzyme UBE2L3 (aka UbcH7), and finally onto the HECT E3 ubiquitin ligase. This assay allowed for the direct comparison of wild-type to the I827K Angelman substituted E6AP$^{\text{HECT}}$ and E6AP$^{\text{C-lobe}}$ activities. As shown in Fig 2, the progression of the ubiquitylation cascade as ubiquitin is transferred sequentially onto the E1 activating enzyme UBE1 in an ATP-dependent manner, followed by a transfer onto the E2 conjugating enzyme UBE2L3. Furthermore, the labile nature of the thiolester UBE1~ubiquitin and UBE2L3~ubiquitin complexes was demonstrated by the addition of the reducing agent DTT (Fig 2). Polyubiquitin chains formation by wild-type E6AP$^{\text{HECT}}$ decreased in the presence of DTT. It is noteworthy that the addition of DTT did not result in a complete reduction of the E3~ubiquitin thioester bond in the E6AP$^{\text{HECT}}$, as was expected, possibly due to non-specific autoubiquitylation of the E6AP$^{\text{HECT}}$

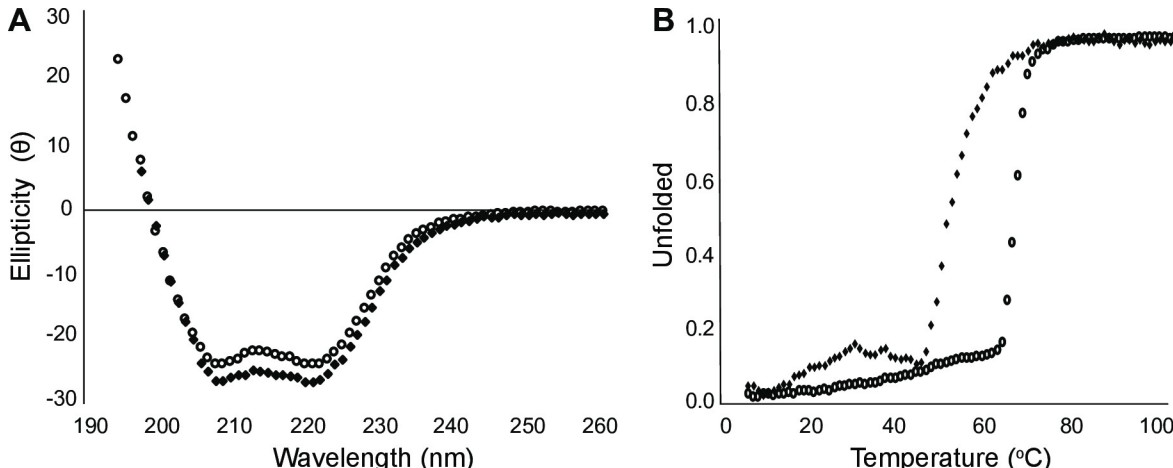

**Fig 1. Circular dichroism spectra for E6AP$^{\text{C-lobe}}$ (wild-type—○, I827K substitution—◆).** A) Wavelength scans of the E6AP$^{\text{C-lobe}}$ constructs show similar secondary structure content at 10˚C. B) Melting curves show the E6AP$^{\text{C-lobe}}$-I827K substitution has a lower the Tm (47.7˚C) than wild-type E6AP$^{\text{C-lobe}}$ (57.7˚C).

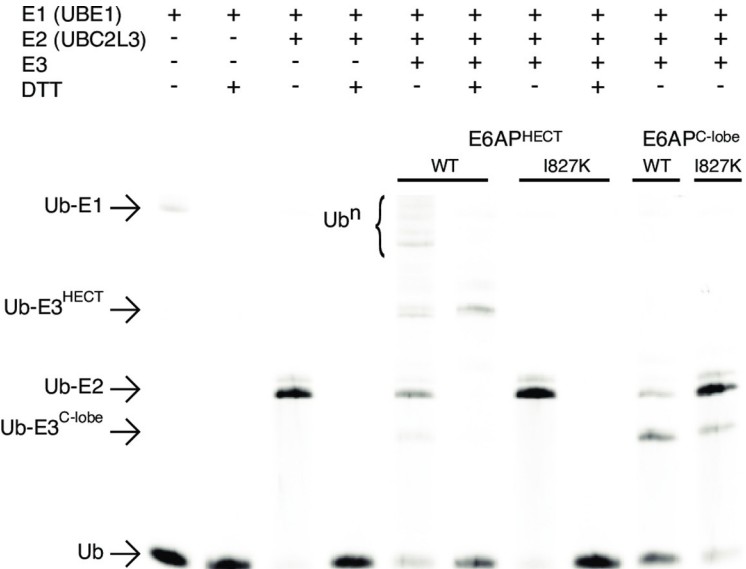

**Fig 2. E6AP I827K is a loss of function substitution.** The ubiquitylation activity assay of wildtype or I827K substituted E6AP$^{HECT}$ and E6AP$^{C-lobe}$ consisted of a 30-minute incubation at 37°C of various combinations of 10 μM ubiquitin N-terminally tagged with Alexa Fluor 647, 10 μM E1 ubiquitin activating enzyme (UBE1), 10 μM E2 ubiquitin conjugating enzyme (UBE2L3), and 35 μM E3 ubiquitin ligase in a buffer containing 50 mM HEPES pH 7.5, 100 mM NaCl, 16 mM ATP, and 40 mM MgCl$_2$. 10 mM DTT was added to alternate samples to reduce any thioester bonds present. The samples were run by electrophoresis a 15% Bis-Tris polyacrylamide gel and visualized on a ThermoFisher iBright FL1000 imaging system.

during the reaction. This is consistent with previous reports of the isolated E6AP$^{HECT}$ being able to autoubiquitylate itself in the absence of substrate [58, 59]. Interestingly, the I827K substitution on the other hand did not result in any polyubiquitin chain formation, demonstrating that the AS substitution decreases ubiquitylation activity. With regards to the E6AP$^{C-lobe}$, the AS I827K mutation also showed diminished activity compared to the wild-type E6AP$^{C-lobe}$. Polyubiquitin chains were not formed in either of the reactions, indicating that the N-lobe is required for the efficient catalysis of chain formation, consistent with activity assays for E6AP$^{C-lobe}$ [59] and other isolated HECT E3 ubiquitin ligase C-lobes for HUWE1, Smurf2, and UBR5 [60, 61]. Interestingly, the AS I827K substituted E6AP$^{C-lobe}$ was still able to be monoubiquitinated, albeit to a lesser extent that wild-type E6AP$^{C-lobe}$. This observation could possibly be due to the structural disruption of the I827K mutation, which prevents the proper presentation of the E6AP$^{C-lobe}$ catalytic cysteine to E2~ubiquitin complex in the absence of the E6AP N-lobe. Mutating the catalytic cysteine to an alanine (C843A) resulted in the complete loss of ubiquitylation activity, as would be expected due to the requirement of E6AP C843 in thiolester bond formation with ubiquitin.

## NMR structural analysis of E6AP$^{C-lobe}$

The $^1$H-$^{15}$N heteronuclear single quantum correlation (HSQC) spectra for $^{15}$N-labeled wild-type E6AP$^{C-lobe}$ and the E6AP$^{C-lobe}$-I827K substituted protein were used to analyze possible structural changes caused by the AS mutation. The wild type E6AP$^{C-lobe}$ spectrum showed a well-folded protein, with well dispersed peaks, indicating that each amino acid was located in its own unique chemical environment (Fig 3A). In contrast, the spectrum of the AS I827K substituted E6AP$^{C-lobe}$ showed many collapsed amide peaks that were mostly localized to the center of the spectrum (Fig 3B) indicating that the protein was partially unfolded. If the protein

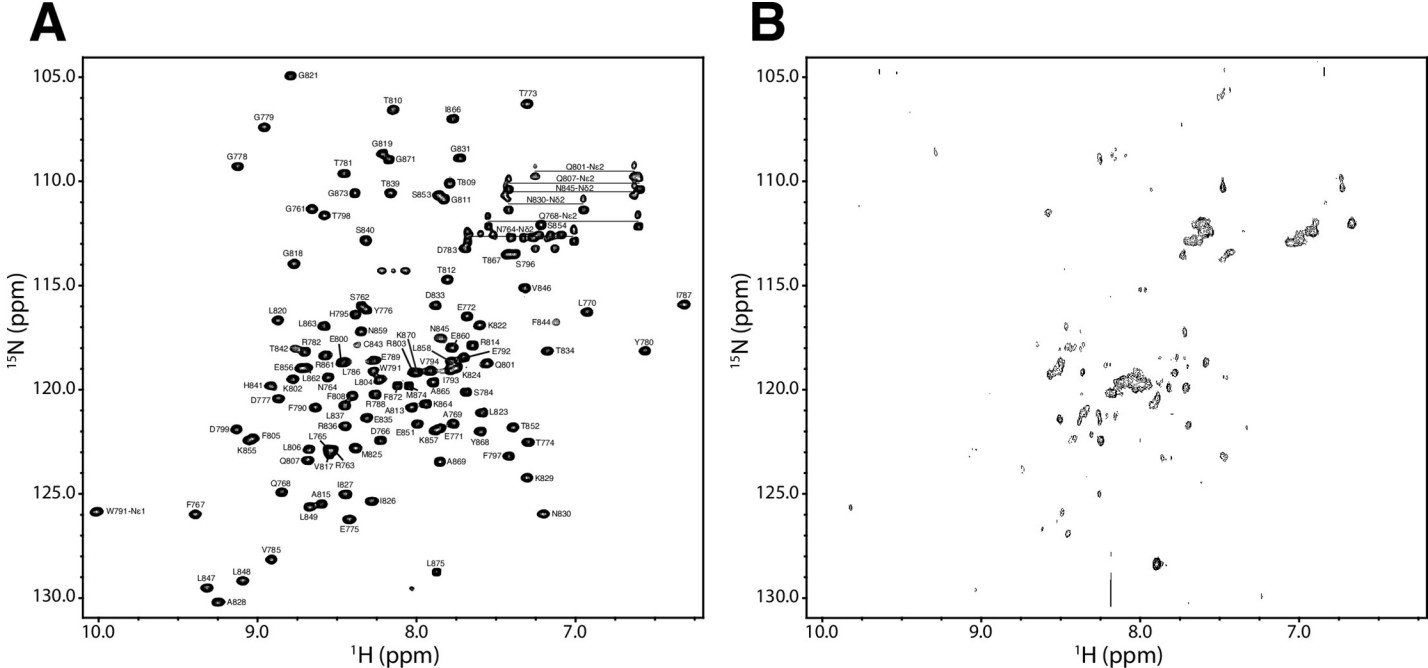

**Fig 3. The I827K substitution destabilizes the 3D fold of E6AP$^{C\text{-lobe}}$ (residues 761–875).** (A) The assigned $^1$H-$^{15}$N-HSQC spectrum of 3.5 mM human E6AP catalytic C-lobe (residues 761–875) using the one-letter amino acid code and residue number according to the human E6AP sequence. The spectrum was determined using standard 3D heteronuclear experiments, and side chain amides for asparagine and glutamine are connected with a horizontal line. (B) The E6AP$^{C\text{-lobe}}$ I827K sample was collected under identical conditions as the wild-type except it was only 188 μM and increased transients collected due to the inherent instability of the protein. The decreased signal intensity and partial collapse amide peaks represent the aggregation of the protein caused to the I827K residue substitution. The NMR samples contained 20 mM Na$_2$HPO$_4$ pH 7.0, 100 mM NaCl, 2 mM TCEP, 1 mM EDTA, and 10% D$_2$O/90% H$_2$O, with imidazole used as an internal pH standard and DSS as the reference point. All data were collected at 25°C on a Varian Inova 600-MHz NMR spectrometer.

was completely unfolded protein would have peaks collapse to the center of the spectrum as solvent exposed peaks no longer experience shielding effects of neighboring amino acids. The disappearance of many peaks can be explained by the decreased signal/noise ratio due to the lower protein concentration as well as the line broadening effects caused by protein aggregation [62]. This is consistent with our inability to purify E6AP$^{C\text{-lobe}}$ I827K protein after multiple attempts using gel filtration chromatography due to the protein aggregating and eluting in the void volume. Furthermore, the concentration used for I827K substituted protein in NMR was similar to the concentrations used in our activity assays and CD experiments, where we did not observe any protein precipitation issues.

Based on the known E6AP crystal structure (PDB 1D5F) [21], we hypothesize that the AS I827K substitution leads to a structural disruption of the E6AP$^{C\text{-lobe}}$ due to a hydrophobic residue being switched to a residue with a positively charged sidechain terminus that would disrupt the hydrophobic network in the core of the domain. The I827 sidechain is not solvent accessible and would not readily accommodate the polar terminus of the lysine sidechain. This is corroborated by the NOESY data that shows the I827 amino acid side chain methyl groups make numerous NOE contacts to several different surrounding hydrophobic side chains including T774, Y776, I787, F790, W791, L824, and M825 (Fig 4). The same rationale would apply to the newly published structure of the domain-swapped E6AP dimer (PDB 6TGK), as I827 is still found embedded in the monomeric hydrophobic core and is not involved in the dimerization interface [63]. This structural disruption likely induces an allosteric change that reduces the catalytic activity and leads to its subsequent aggregation. This is supported by the CD data showing the presence secondary structure at 37°C as well as our observation of

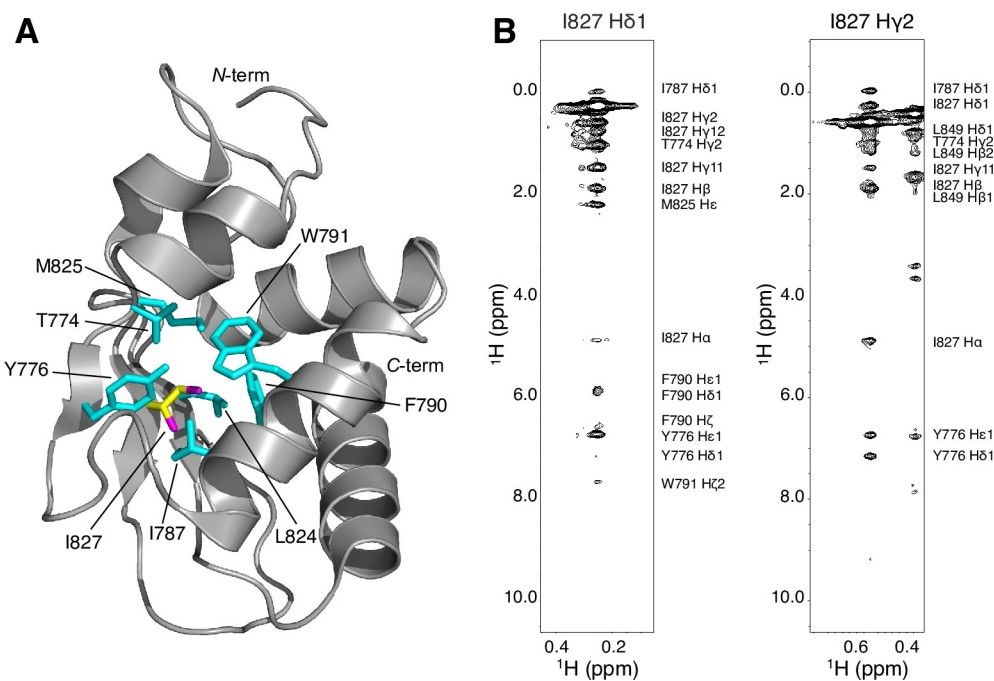

**Fig 4. The I827 residue is an integral residue in the hydrophobic core of E6AP^C-lobe.** (A) The E6AP structure (PDB 1D5F) was rendered in Pymol highlighting the I827 residue (yellow with magenta methyls) surrounded by several aromatic and hydrophobic residues (cyan). (B) Representative $^1$H-$^1$H-NOE strip plots for the I827 methyls showing numerous strong contacts with the surrounding hydrophobic atoms of core residues including T774, Y776, I787, F790, W791, L824, M825, and I827. The authenticity of the assignment was confirmed by reciprocal NOEs from the identified atoms.

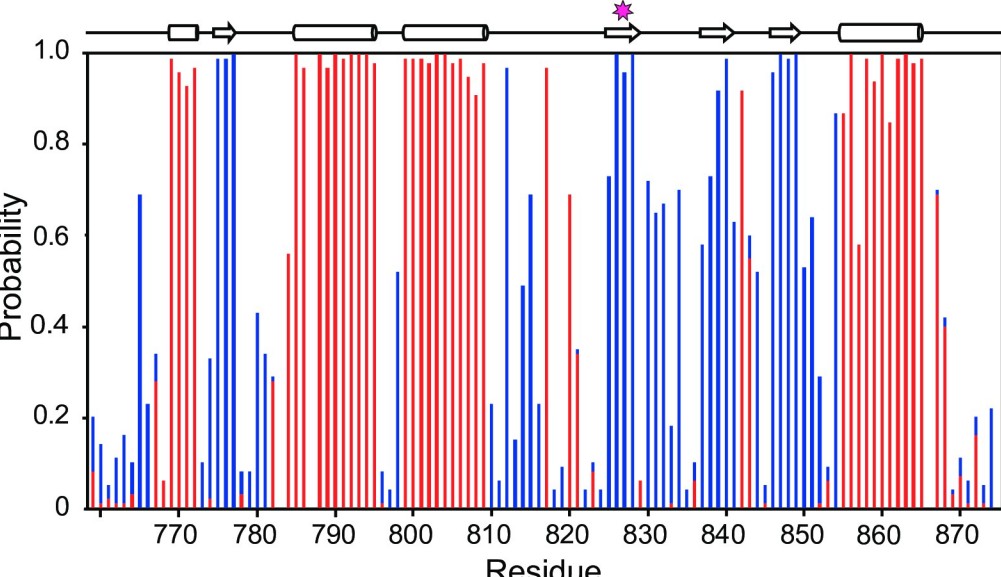

**Fig 5. Predicted secondary structural regions of E6AP^C-lobe.** The probability plot was made by inputting the experimentally determined resonance assignments for E6AP^C-lobe into the online webserver CSI 3.0 [64]. The propensity to form an α-helix or β-strand are denoted in red and blue, respectively. The position of the I827 residue is marked with a star.

aggregated protein eluted in the void volume using gel filtration chromatography. The loss of structural integrity due to the AS I827K substitution correlates with the loss of ubiquitylation activity for E6AP (Fig 2). We are confident that our NMR resonance assignments for the E6AP$^{C-lobe}$ (residues 761–875) are correct and complete as a chemical shift index analysis, which predicts the secondary structure elements based upon chemical shift deviations of the backbone atoms (C$a$, C', C$\beta$, N, H$a$, and NH) [64], are in good agreement with the known tertiary structure of C-terminal lobe of E6AP (Fig 5).

This study provides a structural and biophysical rationale for the E6AP ubiquitin ligase activity loss due to the I827K substitution in AS. This is in good agreement with a previous report that showed the AS I827K substitution resulted in decreased ubiquitylation of the E6AP substrate HHR23A and instability *in vivo* [31]. Continued studies on the E6AP structure and function will help to understand how AS genetic mutations in *UBE3A* result in enzymatic insufficiency and may lead to potential treatments to alleviate the Angelman Syndrome phenotype.

## Supporting information

**S1 Raw images.**
(PDF)

## Acknowledgments

The authors thank Dr. Guoxing Lin for maintaining the 600 MHz NMR spectrometer housed in the Carlson School of Chemistry and Biochemistry at Clark University.

## Author Contributions

**Conceptualization:** Chloe E. Kellum, Rachel J. Orlomoski, Feston Idrizi, Donald E. Spratt.

**Data curation:** Steven A. Beasley, Chloe E. Kellum, Rachel J. Orlomoski, Feston Idrizi, Donald E. Spratt.

**Formal analysis:** Steven A. Beasley, Donald E. Spratt.

**Funding acquisition:** Donald E. Spratt.

**Investigation:** Donald E. Spratt.

**Software:** Steven A. Beasley.

**Supervision:** Donald E. Spratt.

**Validation:** Steven A. Beasley.

**Writing – original draft:** Steven A. Beasley, Chloe E. Kellum, Donald E. Spratt.

**Writing – review & editing:** Steven A. Beasley, Rachel J. Orlomoski, Donald E. Spratt.

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
