## [Decision Letter · Decision Letter 0]

24 Apr 2020

PONE-D-20-07946

An Angelman Syndrome Substitution in the HECT E3 Ubiquitin Ligase C-terminal Lobe of E6AP Affects Protein Stability and Activity

PLOS ONE

Dear Prof. Spratt,

Thank you for submitting your manuscript to PLOS ONE. First, I would sincerely like to apologize for the tardiness in getting this manuscript reviewed. The response to COVID19 has disrupted the schedule of reviewers and this certainly contributed to the slowness. In addition to having one outside reviewer, I also reviewed the manuscript to provide a timely decision for any revision. Below I state my review.

       I agree that I827K mutation has  structural effects on the protein but it is not clear from the HSQC that the C-lobe is completely unfolded. There are still very nicely dispersed signals and the low s/n may be due to both signal broadening due to aggregation and partial unfolding of the protein, and the low concentration of the protein. I wonder if a much longer HSQC of the mutant can be acquired to confirm unfolding. The structure shows that the Ile827 is not completely buried and could accommodate the K mutation. In figure 5, it may be more appropriate to show chemical shift Index rather than the probability plot to confirm that the wt C-lobe is the consistent with the predicted structure.

     In several instances, the concentration of the mutant C-lobe was indicated as 188 or 190 mM (millimolar). I presume you mean micromolar. Also is the concentration of the wt c-lobe indeed 3.5 mM, that seems very high for a domain that requires and make extensive contact with the N-lobe. Was the concentration measured with SUMO attached?

     Can you also provide a statement on whether HECT E3 ligase can poly-autoubiquitinates. Our experience with HECT is that they auto-monoubiquitinate. The fact that the wt- and mutant C-lobe are mono-ubiquitinated suggest that the two protein structures are fairly intact because the interaction with the E2 enzyme is necessary for the ubiquitination. It would also be worthwhile to note precedents that the C-lobe can be ubiquitinated by the E2 enzyme by itself.

With mine and that of the other reviewer, we feel that it has merit but does not fully meet PLOS ONE’s publication criteria as it currently stands. Therefore, we invite you to submit a revised version of the manuscript that addresses the points raised during the review process.

We would appreciate receiving your revised manuscript by Jun 08 2020 11:59PM. To enhance the reproducibility of your results, we recommend that if applicable you deposit your laboratory protocols in protocols.io, where a protocol can be assigned its own identifier (DOI) such that it can be cited independently in the future. For instructions see: http://journals.plos.org/plosone/s/submission-guidelines#loc-laboratory-protocols

We look forward to receiving your revised manuscript.

Kind regards,

Michael Massiah

Academic Editor

PLOS ONE

Reviewers' comments:

Reviewer's Responses to Questions

**Comments to the Author**

1. Is the manuscript technically sound, and do the data support the conclusions?

Reviewer #1: Partly

2. Has the statistical analysis been performed appropriately and rigorously? 

Reviewer #1: N/A

3. Have the authors made all data underlying the findings in their manuscript fully available?

Reviewer #1: Yes

4. Is the manuscript presented in an intelligible fashion and written in standard English?

Reviewer #1: Yes

5. Review Comments to the Author

Reviewer #1: The ubiqutination is one of the most important post-translational modification responsible for protein degradation. The E3A ligase, coded the E6AP protein belonged to HECT ligase family. An Angelman Syndrome observed due to mutation in C-terminal part of HECT domain (C-lobe). The I827K substitution one of the possible mutation leaded to decrease ubiquitin thioester formation and destroy the protein degradation pathway.

The analyzed construct E6AP(C-lobe) analyzed in manuscript comprises residues 761-875 (115 a.a. molecular mass 13 kDa), which is suitable for studies with CD and NMR spectroscopy. For the wild-type E6AP(C-lobe) authors provided high-quality experimental data including 3D 13C and 15N NOESY spectra (Figure 4), which can be used for evaluation of high-resolution 3D structure of the E6AP(C-lobe). However, experimental conditions deserve some criticism, first of all due to the pH of buffer used in experiments close to the isoelectric point (estimated pI 6.94).

As demonstrated by authors, the mutation I827K lead to structural changes, which are clearly visible with CD and NMR techniques. Nevertheless, I couldn’t exclude that observed data are artifact due to experimental conditions. The 15N HSQC spectrum for E6AP(C-lobe) I827K mutant (Figure 3B) suggests a strong aggregation phenomenon rather than unfolded protein. To discriminate between these hypotheses, the authors would have to perform additional experiments.

The short run of structural minimization after I827K substitution shows visible changes in orientation of side-chains of hydrophobic residues, but such dramatic differences were not confirmed experimentally. I could note that I827K mutation increases the pI up to 8.03, which is still very close to pH buffer used in experiments. Further, the K827 together with K829 evidently forms a hydrophilic cluster characterized by positive charge which can destabilize 3D structure of the E6AP(C-lobe). The computer simulations can help understanding that process in details. It’s not surprising that the mutation can facilitate allosteric effects on catalytic C843 resulting in decreased interaction with the polyubiquitination chain.

Summarizing. To discriminate between unfolded or molten globule state, there are some additional experiments required to extract the size of protein in solution, which would help understanding structural changes under I827K mutation. First, we suggest to measure translation diffusion with NMR spectroscopy (DOSY package). To my knowledge, Varian Inova 600 spectrometer is equipped with Performa IV z-gradient unit generated up to 70 G/cm. This is enough to measure diffusion for 13 kDa protein. On the other hand, other experiments can be used, such as the DLS measurements, to estimate size of particles in solution.

Minor points. Authors have to be more careful about manuscript preparation. The experimental conditions for the NMR experiments are different in ‘Materials and Methods’ and in Caption to Figure 3. Also the Figure 4A contains a wrong number of W827 (should be W791).

6. PLOS authors have the option to publish the peer review history of their article (what does this mean?). If published, this will include your full peer review and any attached files.

Reviewer #1: No

---

## [Author Response · Author response to Decision Letter 0]

8 Jun 2020

5 June 2020

Dr. Michael Massiah

Dear Dr. Massiah,

Thank you for the opportunity to resubmit our revised manuscript titled “An Angelman Syndrome Substitution in the HECT E3 Ubiquitin Ligase C-terminal Lobe of E6AP Affect Protein Stability and Activity” that we originally submitted to PLOS ONE on March 19th, 2020. Please find our responses to yours and the reviewer’s specific comments below:

4 April 2020 

PONE-D-20-07946

An Angelman Syndrome Substitution in the HECT E3 Ubiquitin Ligase C-terminal Lobe of E6AP Affects Protein Stability and Activity

PLOS ONE

Dear Prof. Spratt,

Thank you for submitting your manuscript to PLOS ONE. First, I would sincerely like to apologize for the tardiness in getting this manuscript reviewed. The response to COVID19 has disrupted the schedule of reviewers and this certainly contributed to the slowness. In addition to having one outside reviewer, I also reviewed the manuscript to provide a timely decision for any revision. Below I state my review.

Authors’ response: We thank the reviewer and you for reviewing our manuscript during these difficult times. Please excuse our delayed response to yours and the reviewer’s comments – Clark University operations were dramatically affected by the COVID-19 pandemic. We appreciate all of your helpful comments to improve our manuscript and we have addressed and/or incorporated many of your suggested edits.

I agree that I827K mutation has structural effects on the protein but it is not clear from the HSQC that the C-lobe is completely unfolded. There are still very nicely dispersed signals and the low s/n may be due to both signal broadening due to aggregation and partial unfolding of the protein, and the low concentration of the protein. I wonder if a much longer HSQC of the mutant can be acquired to confirm unfolding. The structure shows that the Ile827 is not completely buried and could accommodate the K mutation. 

Author’s response: Thank you for this constructive feedback. Upon further examination of our data, we agree with you that the C-lobe is not completely unfolded. To address this, we have modified the language in the manuscript to state that the I827K Angelman Syndrome substitution causes the partial unfolding and aggregation of the protein. This is in good agreement with our CD data that shows the protein is still folded at temperatures up to the lowered Tm for the mutant. We also observed in our previous gel filtration elution profiles that a significant amount of the E6AP C-lobe I827K protein eluted in void volume indicating that the protein aggregated during purification. The original 1H-15N HSQC that we collected with 64 transients (8x longer than E6AP C-lobe wild-type) was due to significant line broadening that we can attribute to slower protein tumbling caused by protein aggregation. Using Pymol, we observe that the I827 residue is completely buried in the hydrophobic core and is not solvent accessible in the structure (PDB 1D5F), which is in good agreement with the NOE strips shown in Figure 4B. We do not think that the charged ε-amine of the lysine would be easily accommodated within the hydrophobic core, but it does result in the partial unfolding and aggregation of the E6AP C-lobe I827K protein.

In figure 5, it may be more appropriate to show chemical shift Index rather than the probability plot to confirm that the wt C-lobe is the consistent with the predicted structure.

Author’s response: Thank you for this suggestion. The software CSI 3.0 created by Dr. Wishart’s group uses the latest algorithms that incorporates the chemical shift index. We feel the probability plot is appropriate for this figure and would like to keep it as is. The predicted secondary structure boundaries in our CSI 3.0 plot are also in good agreement with the solved E6AP structure (PDB 1D5F) shown in Figure 4. 

In several instances, the concentration of the mutant C-lobe was indicated as 188 or 190 mM (millimolar). I presume you mean micromolar. 

Author’s response: Thank you for pointing out these typos. The concentration of the mutant C-lobe was indeed in micromolar (μM) and the corrections have been made in the manuscript. The discrepancy of 188 or 190 μM was due to a rounding decision made by different authors that worked on the manuscript. This value has been changed in the manuscript to 188 μM for consistency. 

Also is the concentration of the wt c-lobe indeed 3.5 mM, that seems very high for a domain that requires and make extensive contact with the N-lobe. Was the concentration measured with SUMO attached?

Author’s response: The HECT C-lobe is distinct from the N-lobe in all structures of the HECT solved thus far, with the ability to pivot along a small tether. In our experience in working on some of the 28 human HECT constructs, but certainly not all, we have been able to achieve very high concentrations typically in the millimolar (mM) range. For example, in our recent publication on the ITCH C-lobe resonance assignments (Beasley et al., 2019) the 13C-15N enriched recombinant protein was concentrated to 2.6 mM. The His6- or His6-SUMO tag was removed from the constructs prior to NMR data collection and the protein concentrations were determined in the absence of the N-terminal affinity tag.

Can you also provide a statement on whether HECT E3 ligase can poly-autoubiquitinates. Our experience with HECT is that they auto-monoubiquitinate. The fact that the wt- and mutant C-lobe are mono-ubiquitinated suggest that the two protein structures are fairly intact because the interaction with the E2 enzyme is necessary for the ubiquitination. It would also be worthwhile to note precedents that the C-lobe can be ubiquitinated by the E2 enzyme by itself.

Author’s response: Thank you for this helpful suggestion. We have modified the text and expanded our discussion regarding this in our manuscript and included citations for previous reports of the E6AP HECT domain’s autoubiquityation activity. We have also included text referencing previous activity assays for the isolated HECT C-lobes of E6AP, HUWE1, Smurf2, and UBR5. 

With mine and that of the other reviewer, we feel that it has merit but does not fully meet PLOS ONE’s publication criteria as it currently stands. Therefore, we invite you to submit a revised version of the manuscript that addresses the points raised during the review process.

We would appreciate receiving your revised manuscript by Jun 08 2020 11:59PM. To enhance the reproducibility of your results, we recommend that if applicable you deposit your laboratory protocols in protocols.io, where a protocol can be assigned its own identifier (DOI) such that it can be cited independently in the future. For instructions see: http://journals.plos.org/plosone/s/submission-guidelines#loc-laboratory-protocols

 A rebuttal letter that responds to each point raised by the academic editor and reviewer(s). This letter should be uploaded as separate file and labeled 'Response to Reviewers'.

 A marked-up copy of your manuscript that highlights changes made to the original version. This file should be uploaded as separate file and labeled 'Revised Manuscript with Track Changes'.

 An unmarked version of your revised paper without tracked changes. This file should be uploaded as separate file and labeled 'Manuscript'.

Author’s response: The three files requested ('Response to Reviewers', 'Revised Manuscript with Track Changes', and 'Manuscript') have been uploaded to the PLOS ONE editorial manager website. We have also uploaded the revised Figure4.pdf with the corrected residue labeling as noted by Reviewer 1. 

We look forward to receiving your revised manuscript.

Kind regards,

Michael Massiah

Academic Editor

PLOS ONE

Authors response: As requested, the original gel image was uploaded as supplemental information "S1_raw_images". A description for how the image was taken is included in the file.

Author’s response: We have removed the statement as it is not a core part of the research presented in our manuscript.

Reviewer #1’s comments:

The ubiqutination is one of the most important post-translational modification responsible for protein degradation. The E3A ligase, coded the E6AP protein belonged to HECT ligase family. An Angelman Syndrome observed due to mutation in C-terminal part of HECT domain (C-lobe). The I827K substitution one of the possible mutation leaded to decrease ubiquitin thioester formation and destroy the protein degradation pathway.

The analyzed construct E6AP(C-lobe) analyzed in manuscript comprises residues 761-875 (115 a.a. molecular mass 13 kDa), which is suitable for studies with CD and NMR spectroscopy. For the wild-type E6AP(C-lobe) authors provided high-quality experimental data including 3D 13C and 15N NOESY spectra (Figure 4), which can be used for evaluation of high-resolution 3D structure of the E6AP(C-lobe). 

Author’s response: We thank the reviewer for these kind comments. 

However, experimental conditions deserve some criticism, first of all due to the pH of buffer used in experiments close to the isoelectric point (estimated pI 6.94).

Author’s response: We thank the reviewer for this constructive feedback. We agree – the NMR sample could have been collected in an MES buffer at pH 6.0 buffer, similar to our previous publication on ITCH (Beasley et al., 2019) which would have slowed amide exchange to increase solubility and NMR signal to noise. The CD spectra were collected at pH 7.4 and our activity assays were performed at pH 7.5. We would like to point out that the concentration in our E6AP C-lobe wild-type sample, which was collected at pH 7.0, contained 3.5 mM, suggesting that the pH we used was appropriate for our studies. It is also noteworthy that the theoretical wild-type E6AP HECT domain is 5.06, while the E6AP HECT I827K protein is 5.11, which also showed a disruption in ubiquitylation activity. This suggests that I827K substitution is responsible for the observed aggregation and loss of activity and is likely not due to the pH that we used to conduct our experiments.

As demonstrated by authors, the mutation I827K lead to structural changes, which are clearly visible with CD and NMR techniques. Nevertheless, I couldn’t exclude that observed data are artifact due to experimental conditions. The 15N HSQC spectrum for E6AP(C-lobe) I827K mutant (Figure 3B) suggests a strong aggregation phenomenon rather than unfolded protein. To discriminate between these hypotheses, the authors would have to perform additional experiments.

Author’s response: We thank the review for this constructive feedback. Upon further examination of our data, we agree with you that the C-lobe is not completely unfolded and that our NMR spectra and biochemical assays support the loss of activity due to protein aggregation. To address this, we have modified the language in the manuscript to state that the I827K Angelman Syndrome substitution causes the partial unfolding and aggregation of the protein. We also took another look at our previous gel filtration elution profiles during our initial protein purifications and we noted a significant amount of the E6AP C-lobe I827K protein eluted in void volume indicative of protein aggregation. 

The short run of structural minimization after I827K substitution shows visible changes in orientation of side-chains of hydrophobic residues, but such dramatic differences were not confirmed experimentally. I could note that I827K mutation increases the pI up to 8.03, which is still very close to pH buffer used in experiments. Further, the K827 together with K829 evidently forms a hydrophilic cluster characterized by positive charge which can destabilize 3D structure of the E6AP(C-lobe). The computer simulations can help understanding that process in details. It’s not surprising that the mutation can facilitate allosteric effects on catalytic C843 resulting in decreased interaction with the polyubiquitination chain.

Author’s response: We thank the reviewer for bringing this to our attention. We agree that the hydrophilic polylysine cluster is very interesting observation and is something that we will be looking into further in a future study.

Summarizing. To discriminate between unfolded or molten globule state, there are some additional experiments required to extract the size of protein in solution, which would help understanding structural changes under I827K mutation. First, we suggest to measure translation diffusion with NMR spectroscopy (DOSY package). To my knowledge, Varian Inova 600 spectrometer is equipped with Performa IV z-gradient unit generated up to 70 G/cm. This is enough to measure diffusion for 13 kDa protein. On the other hand, other experiments can be used, such as the DLS measurements, to estimate size of particles in solution.

Author’s response: We thank the reviewer for this helpful suggestion. As stated above, after reexamining our size exclusion gel filtration data during purification of the E6AP C-lobe I827K protein we noted that a significant amount of the protein eluted in the void volume, indicative of aggregation. The line broadening seen in the E6AP C-lobe I827K sample (Figure 3B) also indicates protein aggregation. We also thank the reviewer for their advice of running translation diffusion experiments, but unfortunately, we are unable to collect these experiments at this time due to research being halted at Clark University and the current shutdown of our NMR facility due to the COVID-19 pandemic. 

Minor points. Authors have to be more careful about manuscript preparation. The experimental conditions for the NMR experiments are different in ‘Materials and Methods’ and in Caption to Figure 3. Also the Figure 4A contains a wrong number of W827 (should be W791).

Author’s response: Thank the reviewer for noticing these errors. The buffer was corrected in the caption of Figure 3. We have also fixed the tryptophan label in Figure 4A. 

We are very excited about our studies on how E6AP dysfunction contributes to Angelman’s Syndrome and we hope you share our enthusiasm. Thank you again for considering our manuscript for publication in PLOS ONE.

Sincerely,

Donald E. Spratt, Ph.D.

Assistant Professor of Chemistry & Biochemistry

Gustaf H. Carlson School of Chemistry & Biochemistry

Clark University

---

## [Decision Letter · Decision Letter 1]

25 Jun 2020

An Angelman Syndrome Substitution in the HECT E3 Ubiquitin Ligase C-terminal Lobe of E6AP Affects Protein Stability and Activity

PONE-D-20-07946R1

Dear Dr. Spratt,

We’re pleased to inform you that your manuscript has been judged scientifically suitable for publication and will be formally accepted for publication once it meets all outstanding technical requirements.

Kind regards,

Michael Massiah

Academic Editor

PLOS ONE

Additional Editor Comments (optional):

Reviewers' comments:

Reviewer's Responses to Questions

**Comments to the Author**

1. If the authors have adequately addressed your comments raised in a previous round of review and you feel that this manuscript is now acceptable for publication, you may indicate that here to bypass the “Comments to the Author” section, enter your conflict of interest statement in the “Confidential to Editor” section, and submit your "Accept" recommendation.

Reviewer #1: All comments have been addressed

2. Is the manuscript technically sound, and do the data support the conclusions?

Reviewer #1: Yes

3. Has the statistical analysis been performed appropriately and rigorously? 

Reviewer #1: N/A

4. Have the authors made all data underlying the findings in their manuscript fully available?

Reviewer #1: Yes

5. Is the manuscript presented in an intelligible fashion and written in standard English?

Reviewer #1: Yes

6. Review Comments to the Author

Reviewer #1: Well revised manuscript. The authors answered all questions raised, however, some additional experimental work was not done due to Corona problems. The explanations are given instead.

7. PLOS authors have the option to publish the peer review history of their article (what does this mean?). If published, this will include your full peer review and any attached files.

Reviewer #1: No

---

## [Editor Report · Acceptance letter]

26 Jun 2020

PONE-D-20-07946R1 

An Angelman Syndrome Substitution in the HECT E3 Ubiquitin Ligase C-terminal Lobe of E6AP Affects Protein Stability and Activity 

Dear Dr. Spratt:

I'm pleased to inform you that your manuscript has been deemed suitable for publication in PLOS ONE. Congratulations! Your manuscript is now with our production department. 

Kind regards, 

on behalf of

Dr. Michael Massiah 

Academic Editor

PLOS ONE